# C9orf72 polyPR directly binds to various nuclear transport components

Hamidreza Jafarinia, Erik van der Giessen, Patrick R Onck*

Zernike Institute for Advanced Materials, University of Groningen, Groningen, Netherlands

**Abstract** The disruption of nucleocytoplasmic transport (NCT) is an important mechanism in neurodegenerative diseases. In the case of C9orf72-ALS, trafficking of macromolecules through the nuclear pore complex (NPC) might get frustrated by the binding of C9orf72-translated arginine-containing dipeptide repeat proteins (R-DPRs) to the Kapβ family of nuclear transport receptors. Besides Kapβs, several other types of transport components have been linked to NCT impairments in R-DPR-expressed cells, but the molecular origin of these observations has not been clarified. Here, we adopt a coarse-grained molecular dynamics model at amino acid resolution to study the direct interaction between polyPR, the most toxic DPR, and various nuclear transport components to elucidate the binding mechanisms and provide a complete picture of potential polyPR-mediated NCT defects. We found polyPR to directly bind to several isoforms of the Impα family, CAS (the specific exporter of Impα) and RanGAP. We observe no binding between polyPR and Ran. Longer polyPRs at lower salt concentrations also make contact with RanGEF and NTF2. Analyzing the polyPR contact sites on the transport components reveals that polyPR potentially interferes with RanGTP/RanGDP binding, with nuclear localization signal (NLS)-containing cargoes (cargo-NLS) binding to Impα, with cargo-NLS release from Impα, and with Impα export from the nucleus. The abundance of polyPR-binding sites on multiple transport components combined with the inherent polyPR length dependence makes direct polyPR interference of NCT a potential mechanistic pathway of C9orf72 toxicity.

*For correspondence:
p.r.onck@rug.nl

Competing interest: The authors declare that no competing interests exist.

## eLife assessment

This study provides an **important** starting point for unraveling the molecular basis of the pathological phenotypes of the repeat expansion in the gene associated with open-reading frame 72 in human chromosome 9. The coarse-grained simulation method used by the authors goes beyond the state of the art, investigating a **compelling** number of binding partners. The evidence supporting the claims of the authors is **solid**, although experimental validation of the results would strengthen the major conclusions of the work. The work will be of broad interest to biophysicists and biochemists.

## Introduction

The C9orf72 G4C2 hexanucleotide repeat expansion is the most common genetic mutation in amyotrophic lateral sclerosis (ALS) and frontotemporal dementia (FTD) (*Renton et al., 2011*; *DeJesus-Hernandez et al., 2011*). This expansion can be translated into five types of dipeptide repeat proteins (DPRs): polyPR, polyGR, polyGA, polyGP, and polyPA (*Mori et al., 2013*). The positively charged arginine-containing DPRs (R-DPRs) show the highest levels of toxicity in different cell and animal models (*Mizielinska et al., 2014*; *Jovičić et al., 2015*; *Kwon et al., 2014*; *Zhang et al., 2019*; *Zhang et al., 2018b*; *Wen et al., 2014*; *Boeynaems et al., 2016a*), with polyPR known to be the most toxic

DPR (*Jovičić et al., 2015*; *Wen et al., 2014*; *Lee et al., 2016*). R-DPRs have been linked to a wide variety of cellular defects (*Shi et al., 2017*; *Hayes et al., 2020*; *Balendra and Isaacs, 2018*; *Freibaum et al., 2015*; *Hutten et al., 2020*), but a growing body of evidence suggests that neurodegenerative diseases, including C9orf72 ALS/FTD (C9-ALS/FTD), may be caused by disruption of nucleocytoplasmic transport (NCT) (*Kim and Taylor, 2017*; *Prpar Mihevc et al., 2017*; *Boeynaems et al., 2016b*; *Jovičić et al., 2016*). At the same time, many transport components that play a prominent role in NCT have been identified to function as modifiers of G4C2/DPR toxicity (*Jovičić et al., 2015*; *Boeynaems et al., 2016a*; *Kramer et al., 2018*).

The regulated trafficking of proteins and RNA between the nucleus and cytoplasm occurs through nuclear pore complexes (NPCs) embedded in the nuclear membrane (*Stewart, 2007*; *Strambio-De-Castillia et al., 2010*). The NPC is lined with intrinsically disordered phenylalanine-glycine-rich nucleoporins (FG-Nups) that collectively function as a selective permeability barrier. Small molecules rapidly diffuse through the NPC, but the passage of larger cargoes across the barrier needs to be facilitated by their binding to nuclear transport receptors (NTRs) (*Kalita et al., 2021*; *Popken et al., 2015*; *Timney et al., 2016*). The β-karyopherin (Kapβ) family is the largest class of NTRs and includes both import and export receptors (*O'Reilly et al., 2011*). Another essential regulator of NCT is the GTPase Ran, a small protein bound to guanosine triphosphate (GTP) in the nucleus and to guanosine diphosphate (GDP) in the cytoplasm (*Kalita et al., 2021*). The directionality of NCT is mediated by the RanGTP-RanGDP gradient over the nuclear envelope, which is preserved by the cytoplasmic GTPase-activating protein RanGAP and the nuclear guanine nucleotide exchange factor RanGEF (*Kalita et al., 2021*; *Görlich et al., 2003*).

In the import cycle, importins bind their cargoes directly through a nuclear localization signal (NLS) encoded on a cargo. Importin β1 (Impβ1) can also recruit importin α (Impα) that functions as a cargo-adaptor protein. Impα binds to Impβ1 through its N-terminal importin β-binding (IBB) domain (*Pumroy and Cingolani, 2015*; *Lott and Cingolani, 2011*). The importin-NLS-cargo complex then shuttles to the nucleus. The binding of RanGTP to importin in the nucleus disassembles the importin-NLS-cargo complex and the RanGTP-importin complex is recycled to the cytoplasm. When cargo is bound to Impβ1 via Impα, RanGTP dissociates Impβ1 from the Impα-NLS-cargo. This triggers a competition between the flexible IBB domain of Impα and NLS-cargo for binding to Impα, thus facilitating the dissociation of the NLS-cargo. Nucleoporins such as Nup50/Nup2 also catalyze this process by binding to Impα and accelerating the dissociation rate of the cargo-NLS (*Stewart, 2007*). The specific receptor CAS bound to RanGTP is required to export Impα to the cytoplasm. It has been proposed that CAS first displaces Nup50/Nup2 from Impα after which the RanGTP-CAS-Impα returns to the cytoplasm. The hydrolysis of RanGTP to RanGDP in the cytoplasm by RanGAP disassembles the RanGTP-importin and RanGTP-CAS-Impα complexes (*Stewart, 2007*). RanGDP is transported back to the nucleus by nuclear transport factor 2 (NTF2) where the RanGDP-NTF2 complex dissociates when RanGEF regenerates RanGTP (*Stewart, 2007*). In the export cycle, RanGTP promotes the loading of cargoes with nuclear export signal (NES) to the exportin in the nucleus. The resulting RanGTP-exportin-NES-cargo complex moves to the cytoplasm. Once there the complex is disassembled by RanGAP which hydrolyses RanGTP to RanGDP (*Kalita et al., 2021*).

In a recent study, we have analyzed the binding of polyPR to the Kapβ family of importins and exportins (*Hayes et al., 2020*; *Hutten et al., 2020*; *Nanaura et al., 2021*) by using coarse-grained (CG) molecular dynamics simulations (*Jafarinia et al., 2022*). Depending on its length, polyPR can interact with several cargo-, IBB-, RanGTP-, and FG-Nup-binding sites on the Kapβs (*Jafarinia et al., 2022*). Beside Kapβs, there is evidence for direct binding of Impα isomers with R-DPRs (*Hutten et al., 2020*). Some regulators of the Ran cycle are also affected in R-DPR-mediated toxicity, where R-DPRs have been shown to cause mislocalization and abnormal accumulation of RanGAP (*Ryan et al., 2022*), and mislocalization of Ran and RanGEF in cell culture models (*Jovičić et al., 2015*; *Ryan et al., 2022*; *Zhang et al., 2018a*). RanGAP and RanGEF also appear to be modifiers of R-DPR toxicity in genetic studies (*Jovičić et al., 2015*; *Boeynaems et al., 2016a*; *Lee et al., 2016*). It is not clear, however, whether these effects arise from a direct interaction of R-DPRs with NCT components. The aim of the current work is to extend the findings of *Jafarinia et al., 2022* by investigating the interaction between polyPR and various NCT components by means of CG molecular dynamics computations.

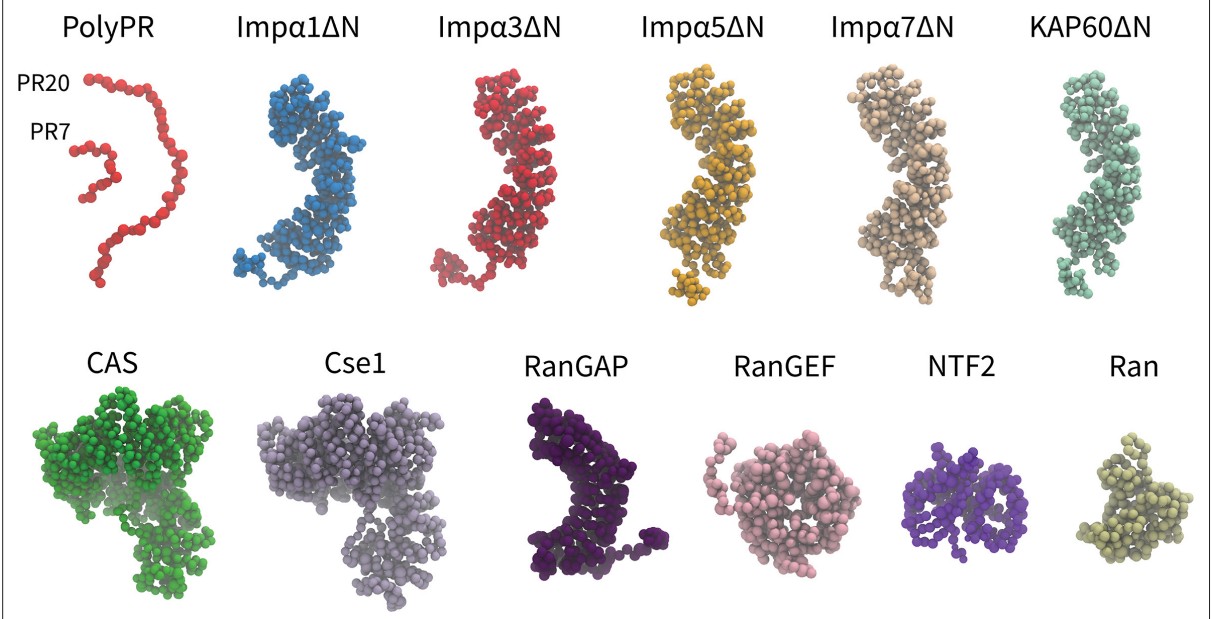

**Figure 1.** Coarse-grained (CG) models of polyPR and transport components used in this study. One-bead-per-amino acid (1BPA) representation of PR7, 20, and the various transport components modeled in the current study. These are several members of the Impα family (excluding the N-terminal IBB domain), specific exporters of Impα (CAS and Cse1), RanGAP, RanGEF, NTF2, and Ran. Details regarding the CG models and protein sequences are listed in **Supplementary file 1b** and **Supplementary file 1c**.

## Results and discussion
### Coarse-grained models of NCT components

We use the residue-scale CG molecular dynamics approach developed and applied earlier to study DPR phase separation (**Jafarinia et al., 2020**) and the direct binding of polyPR to numerous members of the Kapβ family (**Jafarinia et al., 2022**). In the present work, we investigate the interaction of polyPR with unbound human Impα isomers (Impα1, Impα3, Impα5, Impα7), Ran, CAS (the specific exporter of Impα), RanGEF, and NTF2. We also include KAP60 (homolog of Impα), Cse1 (homolog of CAS), and RanGAP from yeast since contrary to the human homologs, the crystal structures of the KAP60-Cse1 complex and the RanGAP-RanGppNHp complex (fission yeast RanGAP bound to the non-hydrolyzable form of human RanGTP) are available in the protein data bank. This enables us to investigate a possible polyPR interference with Impα export and the RanGAP function in the model system of yeast. Moreover, yeast has been employed previously to study NCT defects caused by DPRs (**Jovičić et al., 2015**; **Semmelink et al., 2022**). More details about the selected transport components can be found in **Supplementary file 1b**.

In our one-bead-per-amino acid (1BPA) CG models of transport components, each residue is represented by a single bead at the position of the alpha-carbon atom. The overall tertiary structure of the NCT components is preserved through a network of stiff harmonic bonds, and the distribution of charged and aromatic residues is included in the model. The 1BPA force field correlates with experimental findings for polyPR-Kapβs interactions (**Jafarinia et al., 2022**). The CG models of the NCT components studied are shown in **Figure 1**. The Impα isomers contain a flexible N-terminal IBB domain, followed by a helical core that is constructed from 10 Armadillo (ARM) repeats each consisting of three alpha helices. The NLS-binding sites are located on the concave surface of the helical core (**Pumroy et al., 2015**). The IBB domain has an autoinhibitory role, and when it is not bound to Impβ1, it binds to the ARM concave core and competes with NLS binding (**Stewart, 2007**). To simplify the study of possible binding between polyPR and NLS-binding sites of Impα, the CG models of Impα isomers are built without their N-terminal IBB domains and referred to as ImpαΔN. This approach enables easier investigation of polyPR's interaction with NLS-binding sites. Similar to the importins and exportins, CAS and Cse1 are constructed from HEAT repeats, with each repeat consisting of two antiparallel α-helices, named A and B, connected by linkers of different lengths (**Cook et al.,**

*2005*). RanGEF is mainly constructed from β-strands and has an overall appearance of a seven-bladed propeller with each blade consisting of a four-stranded antiparallel β-sheet (*Renault et al., 1998*). The RanGAP model of fission yeast used in our study is constructed from 11 leucine-rich repeats (LRRs) forming a symmetric crescent followed by a highly negatively charged C-terminal region. Each LRR motif consists of a β-strand-α helix hairpin unit (*Hillig et al., 1999*). NTF2 is a homodimer with each monomer consisting of a β-sheet and three α-helices (*Bullock et al., 1996*). The CG model of Ran is based on the nucleotide-free state of the molecule. More details about the CG models and force field are provided in the 'Methods' section and *Supplementary file 1*.

## PolyPR binds to several transport components through electrostatic interactions

In investigating the direct interaction with the transport components shown in *Figure 1*, we include the potential effect of polyPR length as various studies have found that the repeat length of DPRs strongly correlates with toxicity (*Mizielinska et al., 2014*; *Bennion Callister et al., 2016*; *Swaminathan et al., 2018*; *White et al., 2019*). Simulations of polyPR with varying numbers of repeat units, that is, PR7, PR20, and PR50, are performed at two salt concentrations: 200 mM, similar to previous in vitro experiments performed for Kapβ importins and Impα isomers interacting with R-DPRs (*Hutten et al., 2020*), and a lower ion concentration of 100 mM to study the effect of salt concentration.

To quantify the interaction between polyPR and the transport components, we calculate the time-averaged number of contacts $C_t$ using a cutoff of 1 nm. The number of contacts is normalized by the sequence length of the transport components ($N_{TC}$) and the polyPR length ($N_{PR}$). The normalized number of contacts is plotted against the charge parameter

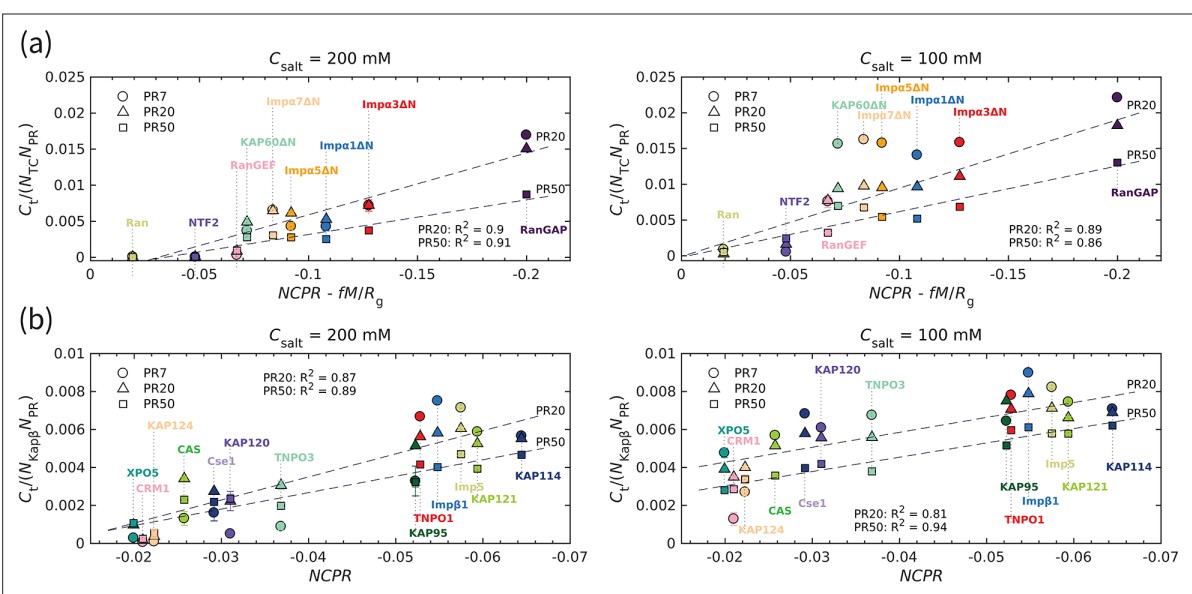

**Figure 2.** The transport component's net charge per residue and dipole moment, together with polyPR length, affect polyPR interaction with various nuclear transport components. (**a**) and (**b**) show the normalized time-averaged number of contacts $C_t$ for the interaction between polyPR with 7, 20, and 50 repeat units with different types of transport components. The results are shown for monovalent salt concentrations of $C_{salt} = 200$ mM (left panels) and $C_{salt} = 100$ mM (right panels). (**a**) shows the results for the transport components shown in *Figure 1*, excluding the specific exporters of Impα: CAS and Cse1. A linear correlation is observed between the normalized $C_t$ and $NCPR - fM/R_g$, with $f$ calculated to be 0.0036 for the best fit. The net charge per residue $NCPR$ is in units of elementary charge e, the dipole moment $M$ is in units of $e \cdot nm$, and the radius of gyration $R_g$ is in units of nm. (**b**) shows the results for the Kapβ data set (data points taken from *Jafarinia et al., 2022*), together with CAS and Cse1. For this case, a linear correlation between the normalized $C_t$ and $NCPR$ is observed. The dashed lines show linear fits for PR20 and PR50; see *Supplementary file 1d* for the linear equations of the fits. The fits for PR7 resulted in $R^2$ values of 0.89 (**a**) and 0.83 (**b**) for 200 M and of 0.7 (**a**) and 0.59 (**b**) for 100 mM. Because of the low $R^2$ values for 100 mM, the fits for PR7 are not shown. The error bars denote the standard error of the mean calculated from block averaging with three blocks at equilibrium. Where error bars are invisible, they are smaller than the marker size.

The online version of this article includes the following figure supplement(s) for figure 2:

**Figure supplement 1.** Quality of fit in *Figure 2*, and corresponding values of dipole moment and net charge per residue of transport components.

$$NCPR - fM/R_g \qquad \qquad (1)$$

where the net charge per residue $NCPR$ is the total charge of the transport component (in units of elementary charge e) divided by its sequence length, $M$ (in units of $e \cdot nm$) is the time-averaged total dipole moment, and $R_g$ (in units of nm) is the time-averaged radius of gyration of the transport components in isolation. The dimensionless parameter $f$ is a free parameter that is calculated to be 0.0036 for the best linear fit in *Figure 2a* for PR20 and PR50, based on the quality of fit ($R^2$) (see *Figure 2—figure supplement 1*). The linear correlation observed in *Figure 2a* confirms an electrostatically driven interaction between polyPR and the transport components. Moreover, it highlights the importance of the spatial distribution of charge over the transport components, as characterized here through the dipole moment. CAS and Cse1, the specific exporters of Impα in human and yeast, respectively, are constructed from HEAT repeats and have a super-helical conformation similar to the Kapβs studied before (*Jafarinia et al., 2022*). We therefore present the results for these two cases jointly with the results for the Kapβ set (taken from *Jafarinia et al., 2022*) in *Figure 2b*. For this set, the best fit is obtained for $f = 0$ (see *Figure 2—figure supplement 1*), which indicates the dominant role of $NCPR$ for the number of contacts between polyPR and the Kapβs, CAS, and Cse1. We attribute this behavior to the structural characteristics of Kapβs, particularly the superhelical structure that features inner and outer surfaces with differing charge distributions. Importantly, this structural arrangement creates an inner surface characterized by a strong negative electrostatic potential. As demonstrated in our previous work, polyPR predominantly binds to this negatively charged cavity within Kapβs. Consequently, the separation of charges on the Kapβ surface becomes less influential compared to the overall charge.

As can be seen in *Figure 2a*, RanGAP features a much larger number of contacts with polyPR than the other transport components, which can be related to the higher negative net charge and the higher dipole moment of this molecule (see the dipole moments $M$ and $NCPR$ of the transport components in *Figure 2—figure supplement 1*). In contrast, polyPR makes a negligible number of contacts with Ran since it has a relatively low dipole moment and no net charge. For NTF2, the value of $NCPR$ is −0.047 e, comparable to several members of the ImpαΔN family, but the lower dipole moment of NTF2 results in a lower number of contacts with polyPR. The binding of PR20 to the ImpαΔN isomers is consistent with the Impα1 and Impα3 binding to R-DPRs found in experiment (*Hutten et al., 2020*), despite the fact that the N-terminal IBB domains of Impα1,3 are excluded from our CG models. At 200 mM salt concentration, polyPR does not bind to NTF2 and Ran. PolyPR contact with RanGEF is also very low at this salt concentration. Reducing the salt concentration to 100 mM increases the number of contacts, clearly indicating that electrostatic force is the main driver for binding. At this lower salt concentration, PR7, PR20, and PR50 make contact with RanGEF, but contact with NTF2 is only observed for longer polyPR chains.

For transport components with higher absolute values of $NCPR$ and $NCPR - fM/R_g$ , the number of contacts increases with increasing polyPR length. However, as can be seen in *Figure 2*, the number of contacts per unit length of polyPR is often seen to be lower for longer polyPRs especially for the lower salt concentrations where polyPR strongly binds to certain transport components; see, for example, the results in *Figure 2a* for the ImpαΔN family and RanGAP at 100 mM salt concentration. This is due to the fact that most of the residues make contact with the target protein for shorter polyPRs, while for longer polyPRs only some parts of the chain are in contact with the transport components and other regions make less or no contact.

## PolyPR interacts with important binding sites of transport components

In order to gain a better understanding of how polyPR interacts with each transport component, we examined the polyPR contact probability of each residue in the sequence of the transport component. The contact probability of each residue is defined as the probability of having at least one polyPR residue within its 1 nm proximity. *Figure 3a* reveals our results of the interaction between PR7 and PR50 with Impα1ΔN, KAP60ΔN, Cse1, RanGAP, RanGEF, and NTF2 (for the other transport components, see *Figure 3—figure supplement 1*). These findings indicate that a longer polyPR makes contact with a larger number of sites and also exhibits a higher contact probability with individual residues compared to a shorter polyPR. We also observe that some regions at the C-terminal ends of the ImpαΔN family and RanGAP are permanently bound to polyPR. We also compare the polyPR-binding sites with known binding sites of transport components (according to the PiSITE webserver

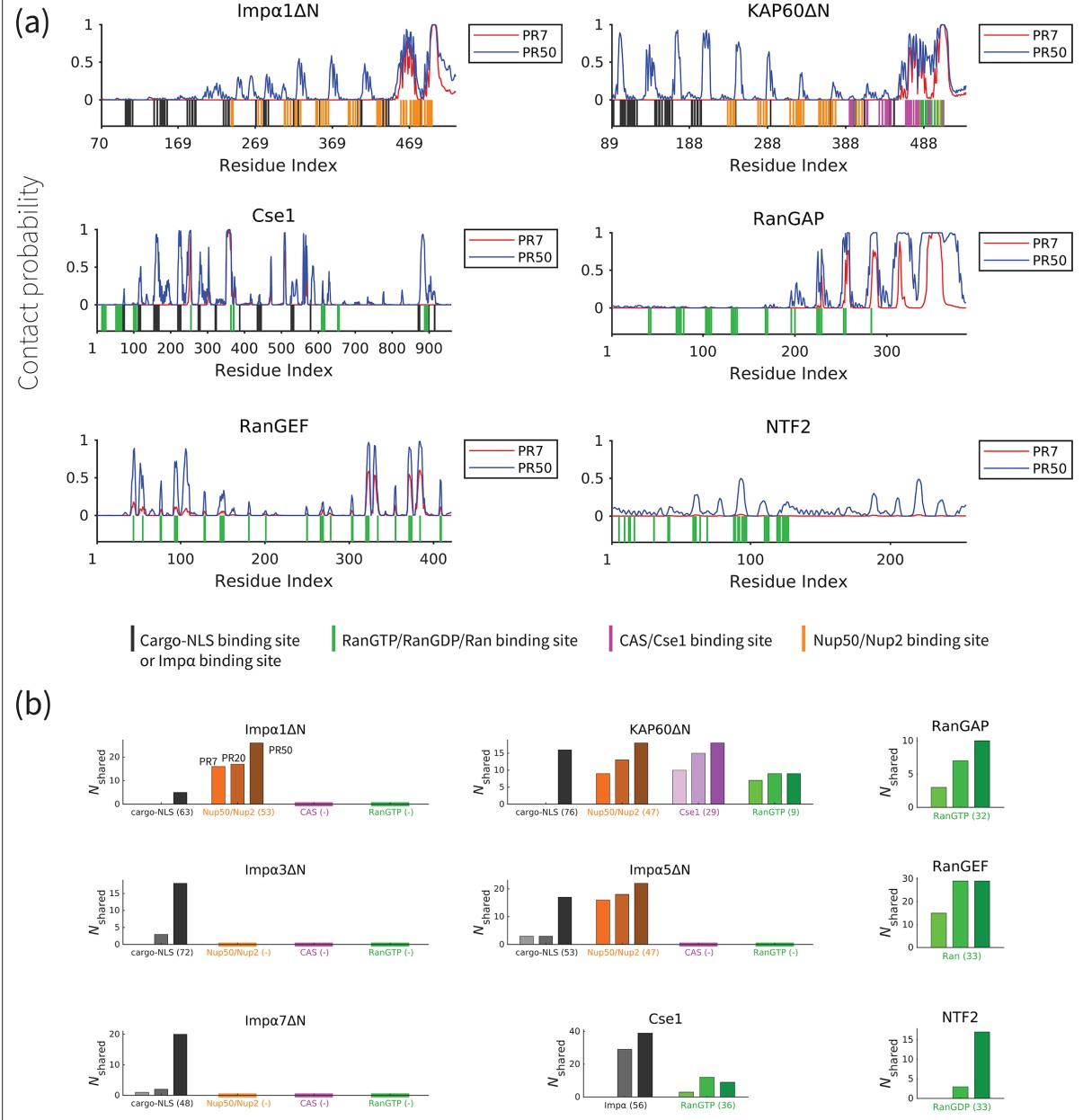

**Figure 3.** PolyPR interacts with several known binding sites of nuclear transport components in a length-dependent manner. (**a**) The contact probability for each residue in the sequence of transport components interacting with polyPR. The plot displays the contact probability for six transport components – Impα1ΔN, KAP60ΔN, Cse1, RanGAP, RanGEF, and NTF2 – at a salt concentration of 100 mM. Results for Impα3ΔN, Impα5ΔN, Impα7ΔN, CAS, and Ran are shown in *Figure 3—figure supplement 1*. Each figure shows two curves for PR7 and PR50. The bottom part of each figure shows the binding sites for NLS-cargo, Impα, CAS/Cse1, RanGTP, and Nup50/Nup2 using different colors. These binding sites are obtained from the crystal structures of the bound states of transport components in the Protein Data Bank using PiSITE (see *Supplementary file 1b* for more details). For each transport component, the following binding sites are marked. For the Impα family: NLS-cargo (vertical black lines), CAS/Cse1 (vertical purple lines), and Nup50/Nup2 (vertical orange lines) binding sites. For CAS/Cse1: Impα (vertical black lines) and RanGTP (vertical green lines) binding sites. For RanGAP: RanGTP (vertical green lines) binding sites. For RanGEF: Ran (vertical green lines) binding sites. For NTF2: RanGDP (vertical green lines) binding sites. The Ran-binding sites marked for RanGEF are taken from the RanGEF-Ran complex (an intermediate step in the RanGEF function). (**b**) The number of shared contact sites between polyPR and the binding partners of the transport components, referred to as $N_{\mathrm{shared}}$, is plotted for PR7, PR20, and PR50. In each bar plot, the numbers inside the parentheses on the horizontal axis show the number of known binding sites obtained from PiSITE. If there is no known binding site, a (-) mark is used instead. The results for PR7, PR20, and PR50 are reported from left to right for each set of bar plots. The bars with darker colors represent longer polyPR chains.

The online version of this article includes the following figure supplement(s) for figure 3:

*Figure 3 continued on next page*

[*Higurashi et al., 2009*]), highlighting them at the bottom of each subfigure. *Figure 3b* displays the number of contact residues shared between polyPR and the native binding partners of each transport component. As expected, the general trend is that the number of shared binding sites increases with increasing polyPR length. PolyPR interacts with the ImpαΔN family at several known cargo-NLS, Nup50/Nup2, and Cse1-binding sites. Longer polyPRs exhibit a significantly stronger interaction with cargo-NLS-binding sites of ImpαΔN compared to shorter polyPRs, which only interact with a limited number of sites. However, we observe that both short and long polyPRs bind to Nup50/Nup2, Cse1, and RanGTP-binding sites, particularly those located near the C-terminal end of ImpαΔN. In the case of Cse1, we observe polyPR binding to Impα and RanGTP-binding sites. PolyPR interacts with the known RanGTP-binding sites of RanGAP. We also show that polyPR is able to bind to the highly negatively charged region in the C-terminal domain of RanGAP that follows the LRR domain. It has been suggested that this negatively charged region is in close proximity to a positively charged region in Ran (in the complex formed by Ran and RanGAP) and plays a role in RanGTP hydrolysis (*Haberland et al., 1997*; *Seewald et al., 2002*). Unfortunately, there is no crystal structure for this region in the PDB structure and thus the binding sites are not known. In the case of RanGEF, a longer polyPR interacts with a high percentage of the known Ran-binding sites. For NTF2, we observe polyPR interaction with RanGDP-binding sites. PolyPR makes negligible contacts with nucleotide-free state of Ran. For CAS, the binding sites are not known. Therefore, these two cases are excluded from *Figure 3b*.

The findings presented in *Figure 3* and *Figure 3—figure supplement 1*, along with previous research on the interaction between polyPR and Kapβs (importins and exportins) (*Jafarinia et al., 2022*), lead to the following mechanistic understanding for the direct effect of polyPR on NCT as illustrated in *Figure 4*. In this figure, the native binding interactions that are affected by polyPR are indicated by red arrows and those that are unaffected by gray arrows. In the import cycle, NLS-cargoes bind to Kapβs directly or indirectly through adaptor proteins such as Impα isomers. PolyPR may impede the loading of cargo to Kapβs and Impα isomers (as shown in inset A) by binding to the cargo-NLS sites. For Impα isomers, the binding of polyPR to cargo-NLS-binding sites is mostly limited to longer chains (see *Figure 3b*). RanGTP disassembles the import complex (cargo-Kapβ or cargo-Impα- Kapβ), and Nup2/Nup50 facilitate the disassembly of the cargo-Impα complex in the nucleus. PolyPR binding to the RanGTP-binding sites on the Kapβs and the Nup2/Nup50-binding sites on Impα (see *Figure 3b*) could result in defects in the dissociation of cargo/cargo-Impα from Kapβ and cargo from Impα (as shown in insets B and C). CAS/Cse1 export Impα by forming a complex with Impα and RanGTP. Findings in *Figure 3b* show that polyPR also interacts with certain CAS/Cse1-binding sites that recognize Impα and RanGTP, possibly interfering with the formation of the RanGTP-CAS/Cse1-Impα complex (as shown in inset D). RanGAP plays a crucial role in the NCT cycle by mediating the hydrolysis of RanGTP to RanGDP, leading to the disassembly of export complexes (RanGTP-importin, RanGTP-CAS/Cse1-Impα, and RanGTP-exportin-cargo). The relatively high number of contacts between polyPR and RanGAP (see *Figure 2a*), as well as polyPR binding to RanGTP-binding sites on RanGAP (see *Figure 3b*), suggests a possible defect in the dissociation of the RanGTP-importin and RanGTP-CAS/Cse1-Impα complexes in the import and Ran cycle, and of the RanGTP-exportin-cargo complex in the export cycle (as shown in inset E). Following the hydrolysis of RanGTP to RanGDP in the cytoplasm, RanGDP is transported back to the nucleus by nuclear transport factor 2 (NTF2). Once in the nucleus, the RanGDP-NTF2 complex dissociates when RanGEF exchanges GDP for GTP in Ran. *Figure 3b* shows that longer polyPRs interact with the Ran-binding sites of RanGEF and NTF2. We therefore suggest that longer polyPRs may also interfere with the Ran cycle by hindering the loading of RanGDP to NTF2 and the exchange of GDP to GTP in Ran by RanGEF (as shown in insets G and F). The interaction between polyPR and NTF2 is distinguished by a relatively lower number of contacts (*Figure 2a*) and contact probabilities for individual NTF2 residues (*Figure 3b*). These findings suggest a lower likelihood of polyPR interfering with NTF2 function compared to the other functions outlined in *Figure 4*. In the export cycle, as examined previously (*Jafarinia et al., 2022*), polyPR binding to RanGTP and FG-Nup-binding sites may affect cargo-loading onto exportins (as shown in inset H) and the transport of exportins through the NPC (as shown in inset I). It should be noted

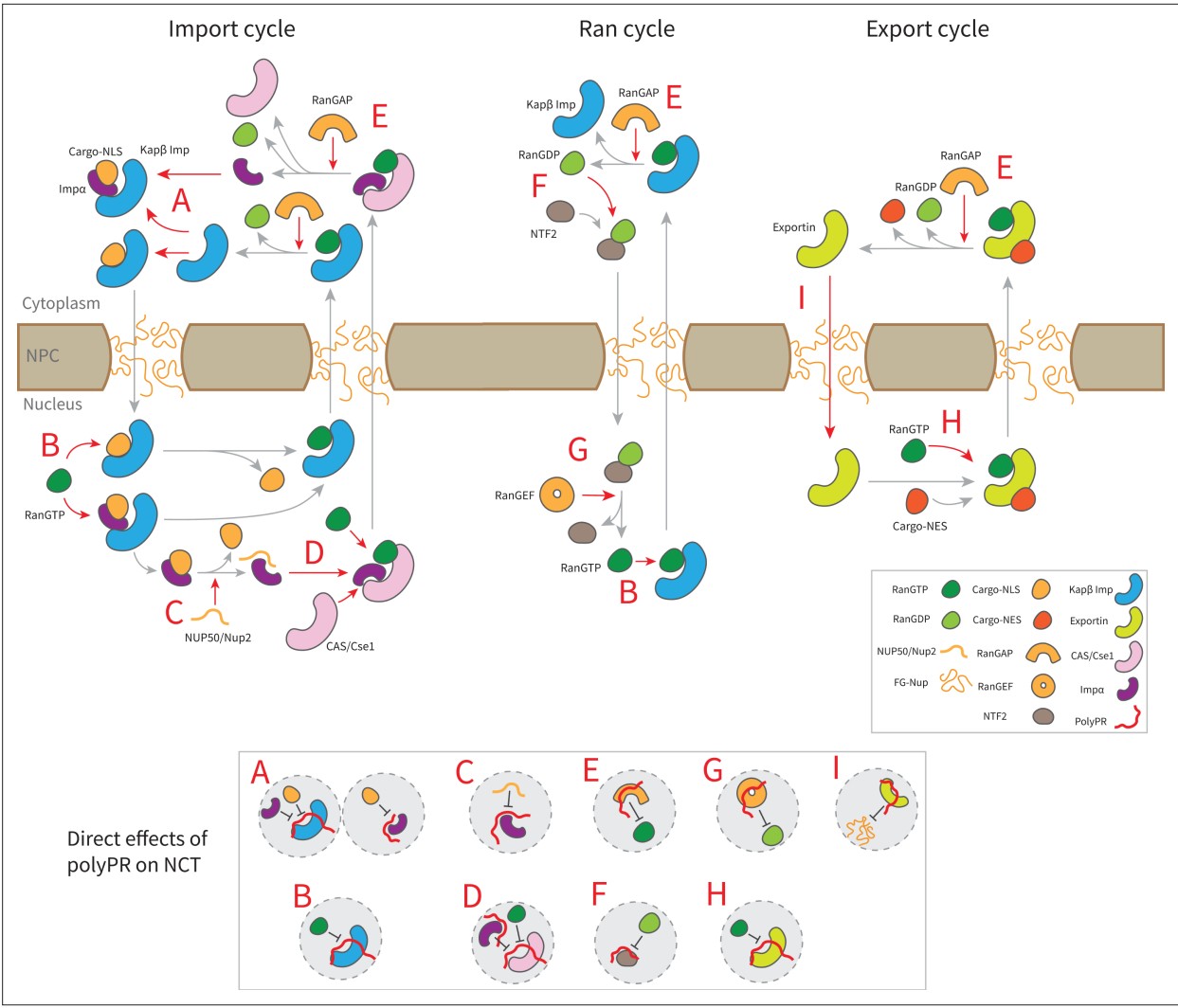

**Figure 4.** Suggested molecular mechanism of polyPR interference with the native function of transport components in the nucleocytoplasmic transport (NCT) cycle. Proposed mechanistic pathways of polyPR interference with the import cycle (left panel), Ran cycle (middle panel), and export cycle (right panel). Steps in the NCT cycle are represented with gray arrows, and a red dashed arrow indicates where polyPR may interfere with the transport cycle. The letters A–H are used to illustrate how polyPR may disrupt the function of the transport components. Each letter corresponds to a mechanistic mechanism shown at the bottom of the figure in gray circles. It should be noted that the proposed mechanisms are not equally significant. The relative significance of the suggested molecular mechanisms can be obtained by considering their relative contributions based on the number of contacts and the number of contacts with important binding sites as presented in *Figures 2 and 3*, respectively.

that the mechanisms proposed in *Figure 4* do not hold equal weight, and the relative contributions based on the number of contacts (presented in *Figure 2*), and the number of contacts with important binding sites (presented in *Figure 3* and *Figure 3—figure supplement 1*) should be considered when comparing the relative significance of the suggested mechanisms.

## Conclusion

In this study, we used CG molecular dynamics simulations to show that polyPR binds to several nuclear transport components. Similar to the interaction of polyPR to the Kapβ family, the interaction of polyPR to other transport components is driven by electrostatic interactions and depends sensitively on polyPR length. Reducing the salt concentration or increasing the polyPR length increases the number of contacts with different transport components. The effect of polyPR length suggests a molecular basis for the more toxic nature of longer polyPRs in animal and cell models (*Mizielinska et al., 2014*; *Bennion Callister et al., 2016*; *Swaminathan et al., 2018*; *White et al., 2019*).

We observed polyPR binding to several members of the Impα family, CAS, Cse1, and RanGAP yet no binding to Ran. PolyPR strongly binds to a highly negatively charged region in the C-terminal domain of fission yeast RanGAP. The human RanGAP also contains a similar highly negatively charged region. Our simulations therefore suggest that a direct interaction may contribute to the observed polyPR-mediated accumulation and mislocalization of RanGAP in HeLa cells (*Ryan et al., 2022*). PolyPR also binds to RanGEF and NTF2 at lower salt concentrations or when the polyPR length is large enough. We also showed that incorporating the dipole moment leads to an improved fit for the transport components analyzed in *Figure 2a*, suggesting that binding is influenced not only by the net charge per residue (as previously observed for Kapβs [*Jafarinia et al., 2022*]) but also by the spatial separation of charges on the transport component.

We showed that polyPR interacts with important binding sites of different transport components in a polyPR-length-dependent manner, with polyPR interaction with RanGTP/RanGDP-binding sites being a common feature between the transport components. This suggests a strong polyPR interference with the Ran gradient across the nuclear envelop. For the ImpαΔN family, we observe polyPR interaction with cargo-NLS and Nup2/Nup50 binding sites. In the case of KAP60ΔN (yeast homolog of Impα), we observe polyPR interaction with Cse1-binding sites. For Cse1 (yeast CAS), we also observed polyPR interacting with Impα-binding sites. These findings suggest polyPR interference with the cargo-NLS association and disassociation with Impα, and the export of Impα.

In conclusion, we showed a pronounced direct binding interaction between polyPR and a surprisingly large number of transport components. By integrating our findings with previously reported data, this work proposes a molecular model that explains how the binding of polyPR might interfere with distinct stages of the transport cycle. The intrinsic length dependence of polyPR binding to important binding sites of many transport components promotes this mechanism to a potential target for therapeutic interventions. Overall, our results offer a basis for future research that aims to explore the impact of C9orf72 R-DPRs on NCT disruption and the subsequent downstream consequences.

## Methods

### Coarse-grained model

We adopted a 1BPA force field to study polyPR interaction with NCT components. This 1BPA approach has been initially developed to simulate disordered FG-Nups (*Popken et al., 2015*; *Ghavami et al., 2014*; *Ananth et al., 2018*; *Fragasso et al., 2021*; *Ghavami et al., 2013*; *Ghavami et al., 2016*; *Dekker et al., 2022*), and extended later to study the phase separation of DPRs (*Jafarinia et al., 2020*), and the interaction of polyPR with Kapβs (*Jafarinia et al., 2022*). The force field potentials and parameters in this study are identical to those employed in *Jafarinia et al., 2022*.

### PolyPR–polyPR interaction

The hydrophobic/hydrophilic interactions between different residues in this force field are represented by

$$\phi_{\text{hp}} = \begin{cases} \varepsilon_{\text{rep}} \left(\dfrac{\sigma}{r}\right)^8 - \varepsilon_{ij} \left[\dfrac{4}{3}\left(\dfrac{\sigma}{r}\right)^6 - \dfrac{1}{3}\right] & r \leq \sigma \\ \left(\varepsilon_{\text{rep}} - \varepsilon_{ij}\right) \left(\dfrac{\sigma}{r}\right)^8 & r \geq \sigma, \end{cases}$$

where $\varepsilon_{ij} = \varepsilon_{\text{hp}} \sqrt{\left(\varepsilon_i \varepsilon_j\right)^{0.27}}$ is the strength of the interaction for each pair of amino acids ($i,j$), $r$ is the distance between beads $i$ and $j$, and $\sigma = 0.6\,\text{nm}$. The values of $\varepsilon_{\text{hp}}$ and $\varepsilon_{\text{rep}}$ are 13 and 10 kJ/mol, respectively. The relative hydrophobic strength values ($\varepsilon_i \in (0, 1)$) of the different amino acids are listed in *Supplementary file 1a*. The hydrophobic strength values of charged residues are slightly increased in line with our recent work (*Jafarinia et al., 2020*). The electrostatic interactions between charged residues are described by the modified Coulomb law:

$$\phi_{\text{elec}} = \frac{q_i q_j}{4\pi\varepsilon_0 \varepsilon_r\left(r\right) r} e^{-\kappa r},$$

where $\varepsilon_r\left(r\right) = S_s \left[1 - \frac{r^2}{z^2}\frac{e^{r/z}}{\left(e^{r/z}-1\right)^2}\right]$ is the distance-dependent dielectric constant of the solvent, with $S_s = 80$ and $z = 0.25\,\text{nm}$. The value of the Debye screening coefficient, $\kappa$, is 1 nm$^{-1}$ for monovalent salt concentration $C_{\text{salt}}$ = 100 mM and 1.5 nm$^{-1}$ for $C_{\text{salt}}$ = 200 mM. For the interactions between the residues within the disordered regions of transport components, we also used the 1BPA force field featuring $\phi_{\text{hp}}$ and $\phi_{\text{elec}}$ as described above.

## PolyPR interaction with transport components

The crystal structure of the transport components is maintained using a stiff harmonic potential $\phi_{\text{network}} = K\left(r - b\right)^2$ , where $K$ is 8000 kJ/mol/nm$^2$ and $b$ is the distance between the amino acid beads in the crystal structure. A bond is made between the beads if $b$ is less than 1.4 nm. The unresolved regions in the crystal structure from the Protein Data Bank and the regions with a lower prediction score (<70 pLDDT) from AlphaFold (*Jumper et al., 2021*; *Varadi et al., 2022*) are included in the CG model as disordered regions. The CG model of fission yeast RanGAP includes two alpha helices in the C-terminal domain, as predicted by AlphaFold. The polyPR interactions with transport components are classified into three categories: (1) electrostatic interactions, (2) cation–pi interactions, and (3) excluded volume interactions. Our force field also accounts for the screening effect of ions.

PolyPR has been shown to bind to several importins in in vitro experiments (*Hutten et al., 2020*). However, no binding has been observed for the more hydrophobic DPRs, that is, polyGA and polyGP (*Hutten et al., 2020*). These observations highlight the importance of arginine in driving the binding between polyPR and NCT components. At physiological salt concentrations, arginine mainly engages in electrostatic and cation–pi interactions. For the polyPR interaction with transport components, we use the same electrostatic potential ($\phi_{\text{elec}}$) as described in the previous section. To take into account the cation–pi interactions between arginine (in polyPR) and the aromatic residues phenylalanine, tyrosine, and tryptophan (in the transport components), we use an 8–6 Lennard–Jones (LJ) potential that replaces $\phi_{\text{hp}}$ for the RF, RY, and RW interactions:

$$\phi_{\text{cp},ij}\left(r\right) = \varepsilon_{\text{cp},ij}\left[3\left(\frac{r_{\text{m}}}{r}\right)^8 - 4\left(\frac{r_{\text{m}}}{r}\right)^6\right],$$

where $r_{\text{m}} = 0.45$ nm is the distance at which the $\phi_{\text{cp},ij}$ reaches its minimum value, and $\varepsilon_{\text{cp},ij}$ is a pair-dependent cation–pi energy taken from *Jafarinia et al., 2022* for different combinations of cation–pi interactions: $\varepsilon_{\text{cp,RF}} = 4.30$, $\varepsilon_{\text{cp,RY}} = 5$, $\varepsilon_{\text{cp,RW}} = 6.7$, $\varepsilon_{\text{cp,KF}} = 1.79$, $\varepsilon_{\text{cp,KY}} = 3.13$ $\varepsilon_{\text{cp,KW}} = 4.26$ kJ/mol. For the hydrophilic/hydrophobic interactions between polyPR and the rest of the transport component residues, including the disordered regions, we use $\phi_{\text{hp}}$ with $\varepsilon_{ij} = 10$ kJ/mol, which leads to an excluded volume potential that vanishes at $r = 0.6$ nm.

## Simulation and analysis

Langevin dynamics simulations were performed at 300 K at monovalent salt concentrations of 100 mM and 200 mM in NVT ensembles with a time step of 0.02 ps and a Langevin friction coefficient of 0.02 ps$^{-1}$ using GROMACS version 2018. Simulations were performed for at least 2.5 μs in cubic periodic boxes, and the last 2 μs were used for analyzing the interaction between polyPR and the transport components. The error bars in *Figure 2* are standard errors of the mean (SEM) calculated from block averaging with three blocks at equilibrium. The binding sites were obtained from the crystal structures of the bound states of transport components in the Protein Data Bank using PiSITE (*Higurashi et al., 2009*). This web-based database provides interaction sites of a protein from multiple PDBs, including similar proteins. The RanGTP binding data (vertical green lines) in *Figure 3* and *Figure 3—figure supplement 1* contains binding residues for both RanGTP and RanGppNHp, the non-hydrolyzable form of RanGTP. The time-averaged number of contacts between the polyPR and transport components in *Figure 2* is obtained by summing the number of contacts per time frame (i.e., the number of polyPR/transport components residue pairs that are within 1 nm) over all frames and dividing by the total number of frames. The contact probability for each transport component residue is the probability of having at least one polyPR residue within 1 nm proximity of the transport component residue. The contact probability is calculated for each transport component residue by dividing the number of frames for which this contact criterion is satisfied by the total number of frames. In

*Figure 3—figure supplement 2*, we show that conducting longer simulations does not significantly affect the contact probabilities presented in *Figure 3a* (data shown for PR50 binding to Impα1ΔN and RanGAP), confirming convergence of our computations. Residue i is considered to be a contact site if the contact probability for this residue is larger than 0.10. $N_{shared}$ is the number of transport component residues that make contact with polyPR (obtained in our simulations) and at the same time are known for recognition of native binding partners (according to PiSITE). The time-averaged total dipole moment of the CG models of transport components was calculated using gmx dipole in GROMACS.

## Additional information

### Funding

| Funder | Grant reference number | Author |
|---|---|---|
| Nederlandse Organisatie voor Wetenschappelijk Onderzoek | Building Blocks of Life | Patrick R Onck |

The funders had no role in study design, data collection and interpretation, or the decision to submit the work for publication.

### Author contributions

Hamidreza Jafarinia, Data curation, Formal analysis, Investigation, Writing - original draft; Erik van der Giessen, Supervision, Methodology, Writing - review and editing; Patrick R Onck, Supervision, Funding acquisition, Methodology, Writing - review and editing

### Author ORCIDs

Hamidreza Jafarinia (iD) http://orcid.org/0000-0003-1205-2978
Erik van der Giessen (iD) http://orcid.org/0000-0002-8369-2254
Patrick R Onck (iD) http://orcid.org/0000-0001-5632-9727

Reviewer #1 (Public Review): https://doi.org/10.7554/eLife.89694.3.sa1
Reviewer #3 (Public Review): https://doi.org/10.7554/eLife.89694.3.sa2
Author Response https://doi.org/10.7554/eLife.89694.3.sa3

## Additional files

### Supplementary files

• Supplementary file 1. Details of the force field parameters, coarse-grained models of transport components, and fitting parameters.

• MDAR checklist

### Data availability

The source data files for all figures in this study can be found in the 'source_data_file.zip' on Dryad at https://doi.org/10.5061/dryad.f7m0cfz46.

The following dataset was generated:

| Author(s) | Year | Dataset title | Dataset URL | Database and Identifier |
|---|---|---|---|---|
| Jafarinia H, Van der Giessen E, Onck P | 2024 | Source data: PolyPR interaction with nuclear transport components | https://datadryad.org/stash/dataset/doi:10.5061/dryad.f7m0cfz46 | Dryad Digital Repository, 10.5061/dryad.f7m0cfz46 |

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
