## [Editor Report · eLife assessment]

This study provides an **important** starting point for unraveling the molecular basis of the pathological phenotypes of the repeat expansion in the gene associated with open-reading frame 72 in human chromosome 9. The coarse-grained simulation method used by the authors goes beyond the state of the art, investigating a **compelling** number of binding partners. The evidence supporting the claims of the authors is **solid**, although experimental validation of the results would strengthen the major conclusions of the work. The work will be of broad interest to biophysicists and biochemists.

---

## [Referee Report · Reviewer #1 (Public Review)]

Jafarinia et al. have made an interesting contribution to unravel the molecular mechanisms underlying pathological phenotypes of repeat expansion of the C9orf72 gene.

The repeat expression leads to expression of polyPR proteins. Using coarse-grained molecular dynamics simulations, the authors identify putative binding partners involved in nucleocytoplasmic transport (NCT), and conjecture that polyPR affects essential processes by binding to NCT-related proteins.

The results are well-reported, but only putative, and need experimental support to be more conclusive. Also, comparison with results from all-atom MD simulations in explicit water could help verify the results. But even without these, the work is very useful as a first step to unravel the role of polyPR and related peptides.

---

## [Referee Report · Reviewer #3 (Public Review)]

Summary:

Onck and co-workers present in this work the identification of binding partners and sites of polyPR on various nuclear transport components and elucidate how polyPR might potentially influence the transport process. It's interesting to note that some interaction sites on transport components also serve as their inherent/functional binding sites (Figure 3). The difference in the effects between short polyPR (PR7) and long polyPR (PR50) is also evident, although the authors might need to clarify the mechanisms better. Overall, I find this manuscript well organized and concisely written, and it would greatly enhance our understanding of the toxicity induced by polyPR.

Strengths:

The 1-bead per atom force field model used in the study is well-tuned for studying the interactions between polyPR and proteins, as the essential cation-pi interactions (between Arg and Phe/Tyr/Trp) was included using a 8-6 LJ model.

Weaknesses:

To cite the author's response: "At the moment, accurately capturing the binding of NCT components to their native binding targets and the competition with polyPR are best resolved by all-atom molecular dynamics simulations, which come with significant computational demands. This level of detail and computation-intensive analyses is beyond the scope of the current study."

---

## [Author Response]

The following is the authors’ response to the original reviews.

We thank the Editor and the referees for their questions and remarks. In this document we provide a point-by-point response to revisions requested by the reviewers.

**Public Reviews:**

**Reviewer #1 (Public Review):**
Jafarinia et al. have made an interesting contribution to unravelling the molecular mechanisms underlying pathological phenotypes of repeat expansion of the C9orf72 gene. The repeat expression leads to the expression of polyPR proteins. Using coarse-grained molecular dynamics simulations, the authors identify putative binding partners involved in nucleocytoplasmic transport (NCT), and that conjecture that polyPR affects essential processes by binding to NCT-related proteins. The results are well-reported, but only putative, and need experimental support to be more conclusive. Also, a comparison with results from all-atom MD simulations in explicit water could help verify the results. But even without these, the work is very useful as a first step to unravel the role of polyPR and related peptides.

We greatly appreciate the reviewer's positive assessment of our work and the suggestions. We acknowledge the need for more experimental validation of the binding behavior of some of the transport components. Our results coincide with the experimental findings of Hutten et al. [1] ([16] in our paper) for example regarding the binding of polyPR to Kapβs and Impαs, but experimental validation of additional transport components, especially for RanGAP, would be valuable. We hope that our work will inspire colleagues from the field to actually perform such experiments.

We also agree with the reviewer's suggestion that all-atom simulations can provide further details on the molecular conformations at the local NTR-PR binding regions. Nonetheless, such simulations for all transport components, particularly for interactions involving large conformational flexibility of longer polyPR chains such as PR50, would require significant computational expenses. In a recent publication (Jafarinia et al. [2]) we reported on the close resemblance in binding behavior between our coarse-grained MD data and the all-atom MD simulations of (Nanaura et al. [3]), both showing polyPR binding to a negatively-charged cavity of Kapβ2. We expect future MD simulations to elucidate more atomistic detail with the continuously increasing power of high-performance computing clusters.

**Reviewer #2 (Public Review):**
This study used coarse-grained molecular dynamics simulation to explain how the binding of polyPR might interfere with distinct stages of the transport cycle. This finding shows that the interaction between polyPR and transport components is driven by electrostatic interactions and is correlated with the salt concentration and the length of polyPR, providing an important basis for subsequent exploration of the impact of C9orf72 R-DPRs on NCT disruption.

We appreciate the reviewer's positive feedback and the recognition of the significance of our work.

**Reviewer #3 (Public Review):**
Onck and co-workers present in this work the identification of binding partners and sites of polyPR on various nuclear transport components and elucidate how polyPR might potentially influence the transport process. It's interesting to note that some interaction sites on transport components also serve as their inherent/functional binding sites. The difference in the effects between short polyPR (PR7) and long polyPR (PR50) is also evident, although the authors might need to clarify the mechanisms better. Overall, the manuscript is well organized and concisely written, and it would greatly enhance our understanding of the toxicity induced by polyPR. In general, the 1-bead per atom force field model used in the study is well-tuned for studying the interactions between polyPR and proteins, as the essential cation-pi interactions (between Arg and Phe/Tyr/Trp) were included using an 8-6 LJ model.

We thank the reviewer for recognizing the suitability of our 1-bead-per-amino-acid force field for studying R-DPRs' interactions with transport components and for acknowledging our work's contribution to understanding polyPR toxicity mechanisms. Below we comment on the mechanisms describing the difference between short and long polyPR molecules.

**Recommendations for the authors:**
1. Regarding Figure 2 (also see below for more specific comments), there is a major concern that the dipole moment is not included in Fig 2b (as the correlation is better with f=0), but the authors still conclude that this is generally important (lines 258-261). As a minimum, this needs to be discussed more carefully. Is f (i..e. the importance of dipole moment for binding) dependent on the specific binding partner, or what is going on? Maybe, there is a good explanation?

Indeed, the significance of the dipole moment depends on the specific type of transport component involved. Our analysis reveals that for Kapβs, see figure 2b, the best-fit is obtained with f=0, indicating that the separation of charge within Kapβs has a relatively minor effect on their interaction with polyPR. Instead, the primary determinant for polyPR-Kapβ interaction appears to be the net charge per residue (NCPR), with a more negative NCPR leading to stronger interactions.

We attribute this behavior to the structural characteristics of Kapβs, particularly the superhelical structure which features inner and outer surfaces with differing charge distributions. Importantly, this structural arrangement creates an inner surface characterized by a negative electrostatic potential. As demonstrated in our previous work, polyPR predominantly binds to this negatively charged cavity within Kapβs. Consequently, the separation of charges on the Kapβ surface becomes less influential compared to the overall charge. Other transport components, however, depicted in figure 2a, do not share this feature and the distribution of charges over the surface becomes a more critical factor in polyPR interactions. We have now added this explanation to page 6, and emphasized in the conclusion section that the effect of dipole moment is only observed for the transport components in figure 2a.

1. Write out nucleoporin, Nup, at first appearance (line 51).

We have changed it in line 51.

1. Fig 1: a (representative) CG structure of polyPR (PR7,PR20 and PR70) would be very useful.

We have added a CG representation of PR7 and PR20 to figure 1.

1. Please use chi-square, not R-square, to evaluate the fit, as chi-square takes experimental errors into account.

We use R-square as a standard measure to assess the quality of the fit in the simulations, as it considers the summation of residuals. This choice aligns with the methodology we have used in our previous publications and therefore prefer to use this measure here as well.

1. Please use a dot (not a full stop) for multiplication in line 151 and Figure 2 legend.

We made the adjustment in line 151, the caption of figure 2, and the y-axis label of figure 2-figure supplement 1.

1. 330: it is very unconventional to plot half the std dev as an error bar. Please plot the std dev (standard error) of the mean.∙

We made the suggested change and now the error bars in figure 2 are standard errors of the mean (SEM) calculated from block averaging with three blocks at equilibrium. We also amended the caption of figure 2 and the Methods section.

1. Please write an explicit equation for the linear relation that is plotted in Figure 2. Something like: C_t = a*(NCPR - f*M/Rg)+b ? That would make it easier to read.

We have now added the linear equation of the fit to a new table Supplementary file 1d, and included a reference to it in the caption of figure 2.

1. Fig 2: why is the fit to PR7 not reported/shown?

The fits for PR7 resulted in R2 values of 0.89 (a) and 0.83 (b) for 200M and of 0.7 (a) and 0.59 (b) for 100 mM. Because of the low R2 values for 100 mM, the fits for PR7 are not shown. We have added this explanation to the caption of figure 2.

1. Fig 4: isn't the blue shape KapB (and not importin)?

We changed "importin" to "Kapβ Imp" for consistency.

1. In the interest of reproducibility, a recommendation is to make the scripts for setting up, running, and analyzing the simulations freely available, e.g. at GitHub. This will increase reproducibility and transparency.

At the moment we do not have the scripts available on GitHub. However, codes can be provided by the authors upon reasonable request, as also mentioned in the data availability statement in the paper.

1. Can the authors explain the salient advances in this article versus the one published last year?

In our previous work, we showed that polyPR binds to the Kapβ family of nuclear transport receptors (NTRs), consistent with experimental findings. While this provided valuable insights, it was essential to broaden our investigation as C9orf72 toxicity not only affects the Kapβ family of NTRs but also disrupts other key regulators of NCT. For instance, recent literature (see lines 87-91 in our paper) showed that Ran and its regulators RanGAP and RanGEF are mislocalized in cells expressing R-DPRs, and genetic screening studies have identified several nucleocytoplasmic transport genes as modifiers of R-DPR-mediated toxicity.

In the present study, we therefore delved deeper into the underlying mechanisms of polyPR-modification of NCT. We focused on exploring whether polyPR directly interacts with Impα isomers, CAS/Cse1, RanGEF, RanGAP, Ran, and NTF2. By doing so, we unveiled a network of direct interactions between polyPR and a remarkably wide range of NCT components. This newfound insight is valuable for interpreting existing experimental findings, such as the mislocalization of RanGAP. We also demonstrate that polyPR binding is influenced not only by factors such as the net charge per residue and the polyPR chain length, as previously observed for Kapβs, but also by the spatial separation of charges, incorporated by an additional dependence on dipole moments in influencing the total number of contacts with polyPR. This sheds new light on how polyPR interacts with numerous targets within the cellular environment, providing a valuable reference for future (experimental) investigations of R-DPR-compromised nuclear transport. These points are explained in the last paragraph of the introduction and paragraphs 2,3 of the conclusion section. Paragraph 2 of the conclusion is also modified for clarification.

1. In Figure 2(a), the vertical coordinates of the first graph do not match the others.

We have now modified figure 2a left panel to match the others.

1. When the polyPR length is large enough, it seems that the binding of polyPR to RanGEF and NTF2 is not significantly improved.

The binding behavior depends on polyPR length, as well as on the net charge per residue and the dipole moment (expressed as NCPR-fM/R_g). We note that the number of contacts in figure 2 is normalized by the polyPR length so that for both NTF2 and RanGEF the total number of contacts increase with length (PR7 to PR20) when binding occurs. Specifically, for RanGEF, especially at lower ion concentrations (100 mM), PR7 and PR20 exhibit a similar number of contacts per unit length of polyPR. This implies that the absolute number of contacts between PR20 and RanGEF is higher than that of PR7. However, as we extend the polyPR length to PR50, there is a reduction in the number of contacts per unit length of polyPR. This phenomenon indicates that the more extended PR50 has regions that make little to no contact with RanGEF, resulting in a smaller number of contacts per unit length for PR50. Lines 188-195 are now modified to put more emphasis on the difference between number of contacts and number of contacts normalized by polyPR length.

1. The representation of the mechanism in Figure 4 is not intuitive enough and the color scheme still needs to be improved.

We have tried to improve clarity by including the names of each transport component next to their schematic representations.

1. Figure 3 shows that the longer polyPR exhibits a higher contact probability with individual residues compared to a shorter polyPR, is this result in conflict with Figure 2?

We re-iterate here that the number of contacts in figure 2 is normalized by the polyPR length, while the results in Fig. 3 are not.

Figure 3 and figure 3-figure supplement 1 demonstrate that as the length of polyPR increases, the contact probability of individual residues of transport components for interaction with polyPR also increases.

In figure 2, we have normalized the time-averaged number of contacts by the length of polyPR. For example, in the top-right panel of figure 2a, when comparing results for PR7 with PR50 interaction with RanGAP, a higher value for PR7 indicates that PR7 makes more contacts per unit of its length with RanGAP. In terms of absolute number of contacts, however, the PR50 chain makes more contacts with RanGAP, resulting in a higher contact probability. We now added a sentence (see lines 188-189) for clarification.

In summary, when a short polyPR strongly binds to a transport component (evidenced by a relatively large number of contacts), it makes more contacts per unit length than a large poyPR. This occurs because for shorter polyPRs most of the residues come into contact with the target protein. In contrast, for longer polyPRs, only certain parts of the chain are in contact with the transport components, while other regions make fewer or no contacts. This is explained in lines 188-195.

1. In S2 and S3, does the data require an error bar?

NCPR, defined as total charge divided by sequence length of the transport components, is a constant and therefore figure 2-figure supplement 1c does not require an error bar.

In figure 2-figure supplement 1b we have added error bars (standard deviation) for the dipole moment calculated from 2.5 us simulations of the isolated transport components.

1. What is the physiological significance when the salt concentration is 100 mM?

We conducted simulations at two different salt concentrations: 200 mM, which aligns with in vitro conditions as reported in Hutten et al. [1], and a lower 100 mM salt concentration. The inclusion of the 100 mM salt concentration enables us to assess the significance of salt concentration, and to confirm the dominance of electrostatic interactions in polyPR binding. We also note that this range of salt concentration is commonly used in in-vitro experiments [1, 4, 5].

1. Please introduce abbreviation NLS in the abstract.

We added the full name of NLS to the abstract.

1. Given the high number of Arg residues in its sequence, polyPR should interact with many proteins. It would be beneficial to discuss the frequency of binding/non-binding interactions of polyPR with nuclear transport components in comparison to general proteins.

We appreciate the reviewer's comment. While such a comparison is indeed interesting, our study primarily focused on elucidating the interactions between polyPR and crucial nuclear transport components, aiming to provide insights into potential defects in nucleocytoplasmic transport. The broader comparison of polyPR interactions with different protein classes in the proteome is indeed an interesting direction for future research, but out of the scope of the current manuscript.

1. The authors should provide a convergence check to determine whether the 2.5 µs simulations are sufficient for sampling the interaction modes, particularly with the long PR50.

We have included a new figure (figure 3-figure supplement 2) and additional text in the Methods section to verify that extending the simulation duration does not alter the contact probabilities (which are indicators of binding modes) presented in figure 3a, confirming convergence of our computations.

1. In reference to Figure 4, the upper panel merely summarizes the known transport mechanisms, while the lower part (A-H) provides potential novel insights from this study. Unfortunately, these novel insights are not sufficiently detailed. It is recommended to include more details to make these relevant plots clearer by expanding the corresponding discussions (currently, only the last paragraph in the Results section addresses these). If possible, the authors should also carry out some CG simulations of the most relevant processes to further elucidate the interference caused by polyPR.

We have taken the reviewer's feedback into consideration and made the suggested revisions. Specifically, we have expanded the last paragraph of the discussion to provide more detailed explanations of the insights derived from our computational model. For each mechanism, we begin by presenting the reader with the baseline understanding of normal function of the transport component. Subsequently, we discuss how the findings presented in figures 2 and 3 offer insights into polyPR's potential interference with the function of NCT components. Furthermore, we have made improvements to the schematic representation of mechanisms in figure 4 to enhance clarity.

At the moment, accurately capturing the binding of NCT components to their native binding targets and the competition with polyPR are best resolved by all-atom molecular dynamics simulations, which come with significant computational demands. This level of detail and computation-intensive analyses is beyond the scope of the current study, but we hope that our results will provide the groundwork for future, more detailed investigations.

**References**

1. Hutten, S., et al., Nuclear Import Receptors Directly Bind to Arginine-Rich Dipeptide Repeat Proteins and Suppress Their Pathological Interactions. Cell Rep., 2020. 33(12): p. 108538.

2. Jafarinia, H., E. Van der Giessen, and P.R. Onck, Molecular basis of C9orf72 poly-PR interference with the β-karyopherin family of nuclear transport receptors. Sci. Rep., 2022. 12(1): p. 21324.

3. Nanaura, H., et al., C9orf72-derived arginine-rich poly-dipeptides impede phase modifiers. Nat Commun, 2021. 12(1): p. 5301.

4. Brady, J.P., et al., Structural and hydrodynamic properties of an intrinsically disordered region of a germ cell-specific protein on phase separation. Proceedings of the National Academy of Sciences, 2017. 114(39): p. E8194-E8203.

5. Fisher, R.S. and S. Elbaum-Garfinkle, Tunable multiphase dynamics of arginine and lysine liquid condensates. Nat. Commun., 2020. 11(1): p. 4628.